# Predictable properties of fitness landscapes induced by adaptational tradeoffs

Suman G Das[1]*, Susana OL Direito[2], Bartlomiej Waclaw[2], Rosalind J Allen[2], Joachim Krug[1]*

[1]Institute for Biological Physics, University of Cologne, Cologne, Germany; [2]School of Physics and Astronomy, University of Edinburgh, Edinburgh, United Kingdom

**Abstract** Fitness effects of mutations depend on environmental parameters. For example, mutations that increase fitness of bacteria at high antibiotic concentration often decrease fitness in the absence of antibiotic, exemplifying a tradeoff between adaptation to environmental extremes. We develop a mathematical model for fitness landscapes generated by such tradeoffs, based on experiments that determine the antibiotic dose-response curves of *Escherichia coli* strains, and previous observations on antibiotic resistance mutations. Our model generates a succession of landscapes with predictable properties as antibiotic concentration is varied. The landscape is nearly smooth at low and high concentrations, but the tradeoff induces a high ruggedness at intermediate antibiotic concentrations. Despite this high ruggedness, however, all the fitness maxima in the landscapes are evolutionarily accessible from the wild type. This implies that selection for antibiotic resistance in multiple mutational steps is relatively facile despite the complexity of the underlying landscape.

*For correspondence:
sdas3@uni-koeln.de (SGD);
jkrug@uni-koeln.de (JK)

**Competing interests:** The authors declare that no competing interests exist.

## Introduction

Sewall Wright introduced the concept of fitness landscapes in 1932 (*Wright, 1932*), and for decades afterwards it persisted chiefly as a metaphor, due to lack of sufficient data. This has changed considerably in recent decades (*de Visser and Krug, 2014*; *Hartl, 2014*; *Kondrashov and Kondrashov, 2015*; *Fragata et al., 2019*). There are now a large number of experimental studies that have constructed fitness landscapes for combinatorial sets of mutations relevant to particular phenotypes, such as the resistance of microbial pathogens to antibiotics (*Weinreich et al., 2006*; *DePristo et al., 2007*; *Marcusson et al., 2009*; *Lozovsky et al., 2009*; *Brown et al., 2010*; *Schenk et al., 2013*; *Goulart et al., 2013*; *Mira et al., 2015*; *Palmer et al., 2015*; *Knopp and Andersson, 2018*), and the genomic scale of these investigations is rapidly growing (*Wu et al., 2016*; *Bank et al., 2016*; *Domingo et al., 2018*; *Pokusaeva et al., 2019*). Mathematical modeling of fitness landscapes has also seen a revival, motivated partly by the need to quantify and interpret the ruggedness of empirical fitness landscapes (*Szendro et al., 2013*; *Weinreich et al., 2013*; *Neidhart et al., 2014*; *Ferretti et al., 2016*; *Blanquart and Bataillon, 2016*; *Crona et al., 2017*; *Hwang et al., 2018*; *Kaznatcheev, 2019*; *Crona, 2020*). Conceptual breakthroughs, such as the notion of sign epistasis (where a mutation is beneficial in some genetic backgrounds but deleterious in others), have shed light on how ruggedness can constrain evolutionary trajectories (*Weinreich et al., 2005*; *Poelwijk et al., 2007*; *Franke et al., 2011*; *Lobkovsky and Koonin, 2012*; *Zagorski et al., 2016*).

Despite this progress, a limitation of current studies of fitness landscapes is that they focus mostly on $G \times G$ (gene-gene) interactions, and little on $G \times G \times E$ (where $E$ stands for environment) interactions, that is on how changes in environment modify gene-gene interactions. A few recent studies have begun to address this question (*Flynn et al., 2013*; *Taute et al., 2014*; *Gorter et al., 2018*;

**eLife digest** Drug resistant bacteria pose a major threat to public health systems all over the world. Darwinian evolution is at the heart of this drug resistance: a mutation that allows bacteria to divide in the presence of a drug appears initially in a single cell. This mutation makes this cell and its descendants more likely to survive, so they can end up taking over the population.

The evolution of resistance can be thought of in terms of 'bacterial fitness landscapes'. These landscapes visualise the relationship between the mutations present in a population of bacteria and how quickly the bacteria divide or reproduce. They are called landscapes because they can be represented as a series of mountains and valleys. The peaks of this landscape represent combinations of mutations that give bacteria the greatest chance of dividing (the greatest fitness). In a landscape with multiple peaks, some peaks will be higher than others. If the landscape is smooth, bacteria can easily acquire mutations for drug resistance. However, in a rugged landscape, bacteria may get stuck at sub-optimal peaks, because the mutations that would enable them to reach a higher peak would first lead them to losing fitness.

Several studies on the evolution of antibiotic resistance exist for specific bacteria and specific drugs, but relatively little is known about the general properties of the underlying fitness landscapes. Do these landscapes have features that can help explain the rapid evolution of high levels of resistance?

Antibiotic resistance often comes at a cost – more resistant strains of bacteria tend to grow more slowly when the drug is absent. To build a model of antibiotic resistance landscapes, Das et al. performed growth experiments on several strains of *Escherichia coli* exposed to a drug called ciprofloxacin. They measured how the rate at which the bacteria divided changed at different antibiotic concentrations, and combined this with the observation about resistant strains growing slower to formulate a mathematical model of antibiotic resistance landscapes. The landscapes that resulted were found to be very rugged, but unexpectedly, the bacteria could still evolve to access all fitness peaks. This means that landscape ruggedness does not constrain the evolution of resistance.

Understanding how and when resistance evolves is important both for the design of new drugs and the development of treatment protocols. A specific prediction of the model is that resistance evolution in fitness landscapes where resistant strains divide more slowly is reversible. This implies that the bacteria could regain their susceptibility to treatment when the drug concentration decreases, but this would depend on the specific bacteria and drug in question. More broadly, the model provides a framework for addressing the evolution of resistance in clinical and environmental settings, where drug concentrations vary widely in time and space.

*de Vos et al., 2018*). In the context of antibiotic resistance, it has been realized that the fitness landscape of resistance genes depends quite strongly on antibiotic concentration (*Mira et al., 2015*; *Stiffler et al., 2015*; *Ogbunugafor et al., 2016*). This is highly relevant to the clinical problem of resistance evolution, since concentration of antibiotics can vary widely in a patient's body as well as in various non-clinical settings (*Kolpin et al., 2004*; *Andersson and Hughes, 2014*). Controlling the evolution of resistance mutants thus requires an understanding of fitness landscapes as a function of antibiotic concentration. Empirical investigations of such scenarios are still limited, and systematic theoretical work on this question is also lacking.

In the present work, we aim to develop a theory of $G \times G \times E$ interactions for a specific class of landscapes, with particular focus on applications to antibiotic resistance. The key feature of the landscapes we study is that every mutation comes with a tradeoff between adaptation to the two extremes of an environmental parameter. For example, it has been known for some time that antibiotic resistance often comes with a fitness cost, such that a bacterium that can tolerate high drug concentrations grows slowly in drug-free conditions (*Andersson and Hughes, 2010*; *Melnyk et al., 2015*). While such tradeoffs are not universal (*Hughes and Andersson, 2017*; *Durão et al., 2018*), they certainly occur for a large number of mutations and a variety of drugs.

Tradeoffs can also arise in complex scenarios involving multiple drugs. It has been reported in *Stiffler et al., 2015* that certain mutations in TEM-1 $\beta$-lactamase are neutral at low ampicillin concentration but deleterious at high concentration, and that a number of the latter mutations also

confer resistance to cefotaxime. Therefore in a medium with cefotaxime and a moderately high concentration of ampicillin, it is possible that these mutations will be deleterious at low cefotaxime concentrations but beneficial at high cefotaxime concentration. Fitness landscapes with adaptational tradeoffs are therefore also of potential relevance to evolution in response to multi-drug combinations.

Our starting point for investigating fitness landscapes induced by tradeoffs is the knowledge of two phenotypes that are well studied – the drug-free growth rate (which we call the null-fitness) and the $IC_{50}$ (the drug concentration that reduces growth rate by half), which is a measure of antibiotic resistance. These two phenotypes correspond to the two extreme regimes of an environmental parameter, that is zero and highly inhibitory antibiotic concentrations. The function that describes the growth rate of a bacterium for antibiotic concentrations between these two extremes is called the dose-response curve or the inhibition curve (*Regoes et al., 2004*). When tradeoffs are present, the dose-response curves of different mutants must intersect as the concentration is varied (*Gullberg et al., 2011*). This is schematically shown in *Figure 1*. The intersection of dose-response curves of the wild type and the mutant happens at point A, swapping the rank order between the two fitness values. The intersection point is known as the minimum selective concentration (MSC), and it defines the lower boundary of the mutant selection window (MSW) within which the resistance mutant has a selective advantage relative to the wild type (*Khan et al., 2017*; *Alexander and MacLean, 2018*).

When there are several possible mutations and multiple combinatorial mutants, a large number of such intersections occur as the concentration of the antibiotic increases. This leads to a succession of different fitness landscapes defined over the space of genotype sequences (*Maynard Smith, 1970*; *Kauffman and Levin, 1987*). Whenever the curves of two mutational neighbors (genotypes that differ by one mutation) intersect, there can be an alteration in the evolutionary trajectory towards resistance, whereby a forward (reverse) mutation now becomes more likely to fix in the population than the corresponding reverse (forward) mutation. These intersections change the ruggedness of landscapes and the accessibility of fitness maxima. In this way a rich and complex structure of selective constraints emerges in the MSW. To explore the evolutionary consequences of these constraints, here we construct a theoretical model based on existing empirical studies as well as our own work on ciprofloxacin resistance in *E. coli*. Specifically, we address two fundamental questions: (i) How does the ruggedness of the fitness landscape vary as a function of antibiotic concentration? (ii) How accessible are the fitness optima as a function of antibiotic concentration?

We find that even when the null-fitness and resistance values of the mutations combine in a simple, multiplicative manner, the intersections of the curves produce a highly epistatic landscape at intermediate concentrations of the antibiotic. This is an example of a strong $G \times G \times E$ interaction, where changes in the environmental variable drastically alter the interactions between genes. Despite the high ruggedness at intermediate concentrations, however, the topology of the landscapes is systematically different from existing oft-studied random landscape models, such as the House-of-Cards model (*Kauffman and Levin, 1987*; *Kingman, 1978*), the Kauffman NK model (*Kauffman and Weinberger, 1989*; *Hwang et al., 2018*) or the Rough Mt. Fuji model (*Neidhart et al., 2014*). For example, most fitness maxima have similar numbers of mutations that depend logarithmically on the antibiotic concentration. Importantly, all the fitness maxima remain highly accessible through adaptive paths with sequentially fixing mutations. In particular, any fitness maximum (including the global maximum) is accessible from the wild type as long as the wild type is viable. As a consequence, the evolution of high levels of antibiotic resistance by multiple mutations (*Hughes and Andersson, 2017*; *Wistrand-Yuen et al., 2018*;

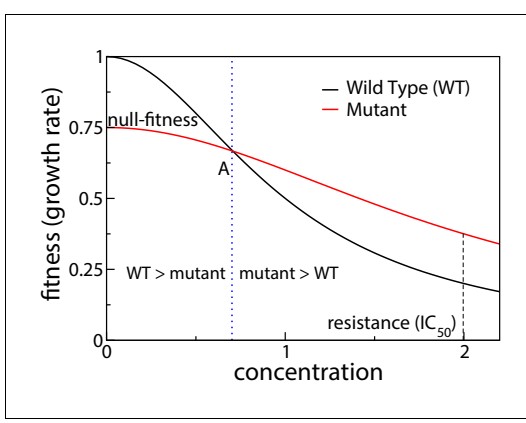

**Figure 1.** Schematic showing dose response curves of a wild type and a mutant. To the left of the intersection point A the wild type is selected over the mutant, whereas to the right of A the mutant is selected.

*Rehman et al., 2019*) is much less constrained by the tradeoff-induced epistatic interactions than might have been expected on the basis of existing models.

## Results

### Mathematical model of tradeoff-induced fitness landscapes

The chief goal of this paper is to develop and explore a mathematical framework to study tradeoff-induced fitness landscapes. We consider a total of $L$ mutations, each of which increases antibiotic resistance. A fitness landscape is a real-valued function defined on the set of $2^L$ genotypes made up of all combinations of these mutations. A genotype can be represented by a binary string of length $L$, where a 1 (0) at each position represents the presence (absence) of a specific mutation. Alternatively, any genotype is uniquely identified as a subset of the $L$ mutations (the wild type is the null subset, that is the subset with no mutations).

In this paper, unless mentioned otherwise, we define the fitness $f$ as the exponential growth rate of a microbial population. The fitness is a function of antibiotic concentration. This function has two parameters of particular interest to us – the growth rate at zero concentration, which we refer to as the null-fitness and denote by $r$, and a measure of resistance such as IC$_{50}$ which we denote by $m$. Each single mutation is described by the pair $(r_i, m_i)$, where $r_i$ and $m_i$ are the null-fitness and resistance values respectively of the $i$th single mutant. We further rescale our units such that for the wild type, $r = 1$ and $m = 1$. We consider mutations that come with a fitness-resistance tradeoff, that is a single mutant has an increased resistance ($m_i > 1$) and a reduced null-fitness ($r_i < 1$) compared to the wild type. To proceed we need to specify two things: (i) how the fitness of the wild type and the mutants depend on antibiotic concentration, and in particular if this dependence exhibits a pattern common to various mutant strains; (ii) how the $r$ and $m$ values of the combinatorial mutants depend on those of the individual mutations. To address these issues we take guidance from two empirical observations.

#### Scaling of dose-response curves

*Marcusson et al., 2009* have constructed a series of *E. coli* strains with single, double and triple mutations conferring resistance to the fluoroquinolone antibiotic ciprofloxacin (CIP), which inhibits DNA replication (*Drlica et al., 2009*). In their study they measured MIC (minimum inhibitory concentration) values and null-fitness but did not report dose-response curves. Some of the present authors have recently shown that the dose-response curve of the wild-type *E. coli* (strain K-12 MG1655) in the presence of ciprofloxacin can be fitted reasonably well by a Hill function (*Ojkic et al., 2019*).

Here we expand on this work and determine dose-response curves for a range of single- and double-mutants with mutations restricted to five specific loci known to confer resistance to CIP (*Marcusson et al., 2009*) (see Materials and methods). *Figure 2A* shows the measured curves for the wild type, the five single mutants, and eight double-mutant combinations. The genotypes are represented as binary strings, where a 1 or 0 at each position denotes respectively the presence or absence of a particular mutation. If we rescale the concentration $c$ of CIP by IC$_{50}$ of the corresponding strain, $x = c/IC_{50}$, and the growth rate by the null-fitness $f(0)$, the curves collapse to a single curve $w(x)$ that can be approximated by the Hill function $(1 + x^4)^{-1}$ (*Figure 2B*). The precise shape of the curve is not important for further analysis in this paper. However, the data collapse suggests that we can assume that the dose-response curve of a mutant with (relative) null-fitness $r$ and (relative) resistance $m$ is

$$f(c) = rw(c/m), \tag{1}$$

that is, it has the same shape as the wild-type curve $w$ except for a rescaling of the fitness and concentration axes. Similar scaling relations have been reported previously by *Wood et al., 2014* and *Chevereau et al., 2015*. A good biological understanding of the conditions underlying this feature is presently lacking, but it seems intuitively plausible that the shape $w(x)$ would be robust to changes that do not qualitatively alter the basic physiology of growth and resistance.

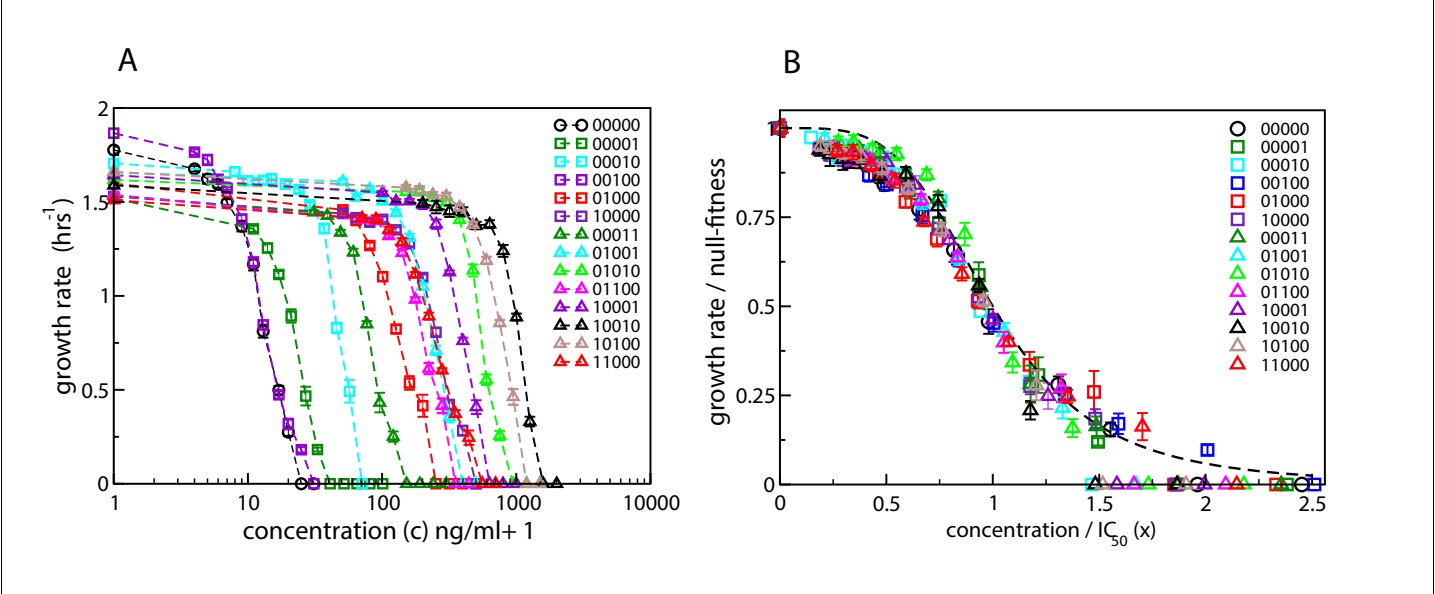

**Figure 2.** Dose-response curves for *E. coli* in the presence of ciprofloxacin. Each binary string corresponds to a strain, where the presence (absence) of a specific mutation in the strain is indicated by a 1 (0). The five mutations in order from left to right are S83L (*gyrA*), D87N (*gyrA*), S80I (*parC*), Δ*marR*, and Δ*acrR*. The names of the strains are given in *Table 1*. (A) Dose-response curves of the wild type, the five single mutants and eight double mutants. Unlike the experiments reported in *Marcusson et al., 2009*, the mutants were grown in isolation rather than in competition with the wild type. (B) The same curves, but scaled with the null-fitness and IC$_{50}$ of each individual genotype. The dashed black line is the Hill function $(1 + x^4)^{-1}$.

## Limited epistasis in $r$ and $m$

An interesting recent finding reported by *Knopp and Andersson, 2018* is that chromosomal resistance mutations in *Salmonella typhimurium* mostly alter the null-fitness as well as the MIC of various antibiotics in a non-epistatic, multiplicative manner, that is if a particular mutation increases (decreases) the resistance (null-fitness) by a factor $k_1$, and another mutation does the same with a factor $k_2$, then the mutations jointly alter these phenotypes roughly by a factor of $k_1 k_2$ (with a few exceptions). We have done a similar comparison for the data on the null-fitness and MIC for *E. coli* strains in *Marcusson et al., 2009*. We have analyzed a subset of 4 mutations for which the complete data set for all combinatorial mutants is available from *Marcusson et al., 2009*. The data are shown in *Table 1*. Out of 11 multiple-mutants, only 2 show epistasis in $r$ and 4 show epistasis in $m$. Moreover, in all cases where significant epistasis occurs it is negative, that is the effect of the multiple mutants is weaker than expected from the single mutation effects.

## Formulation of the model

The above observations suggest a model where one assumes, as an approximation, that all the $r$ and $m$ values of individual mutations combine multiplicatively. A genotype with $n$ mutations $(r_1, m_1), (r_2, m_2), \ldots, (r_n, m_n)$ has a null-fitness $r$ and a resistance value $m$ given by

$$r = \prod_{i=1}^{n} r_i \quad \text{and} \quad m = \prod_{i=1}^{n} m_i. \tag{2}$$

Moreover, the dose-response curves of the genotypes are taken to be of the scaling form (*Equation 1*), where the function $w(x)$ does not depend on the genotype. As indicated before, and without any loss of generality, we choose units such that, for the wild type, $r = 1$ and $m = 1$. Therefore the dose-response curve of the wild type is $w(x)$ with $w(0) = 1$, and choosing IC$_{50}$ as a measure of resistance, we have $w(1) = \frac{1}{2}$. Henceforth, we refer to $x$ simply as the concentration. We also recall that the condition of adaptational tradeoff means that $r_i < 1$ and $m_i > 1$ for all mutations.

If the $r_i$ and $m_i$ values combine non-epistatically, and if the shape of the dose-response curve is known, it is thus possible to construct the entire concentration-dependent landscape of size $2^L$ from

**Table 1.** Epistasis in null-fitness and MIC for *E. coli* in the presence of ciprofloxacin
The table contains a combinatorially complete subset of the data reported by *Marcusson et al., 2009*, composed of the four mutations S83L (gyrA), D87N (gyrA), marR, and acrR. The names of the strains and values of null-fitness (in competition assays with the wild type) in the third column and MIC (of ciprofloxacin) in the fifth column are obtained from *Marcusson et al., 2009*. The binary representations follow the same convention as given in the caption of Figure 2. The fourth and sixth columns are respectively the null-fitness and MIC values expected in the absence of epistasis. NA denotes the cases where this is not applicable. The values in parentheses are error estimates. In the third and fifth columns, the errors in the $\log(x)$ are calculated as $\frac{|\Delta x|}{x}$, where $|\Delta x|$ are the standard error as calculated from the standard deviations reported in the paper. The errors in columns four and six were estimated as $\sum_i \frac{|\Delta x_i|}{x_i}$ where the sum is over the mutations present in the combinatorial mutants. The detectable cases of epistasis are marked in blue. Negative epistasis is found in all these cases. Also, all the cases with epistasis correspond to two or more mutations that affect the same chemical pathways.

| Strain | String | Log null-fitness | Non-epistatic | Log MIC | Non-epistatic |
|--------|--------|------------------|---------------|---------|---------------|
| MG1655 | 00000 | 0.00 (±0.004) | NA | 0.00 (±0.35) | NA |
| LM378 | 10000 | 0.01 (±0.016) | NA | 3.17 (±0.70) | NA |
| LM534 | 01000 | −0.01 (±0.018) | NA | 2.75 (±0.70) | NA |
| LM202 | 00010 | −0.19 (±0.020) | NA | 0.69 (±0.70) | NA |
| LM351 | 00001 | −0.094 (±0.014) | NA | 1.08 (±0.70) | NA |
| LM625 | 11000 | −0.030 (±0.011) | 0.0 (±0.038) | 3.17 (±0.70) | 5.92 (±1.1) |
| LM421 | 10010 | −0.15 (±0.019) | −0.18 (±0.040) | 4.13 (±0.70) | 3.56 (±1.1) |
| LM647 | 10001 | −0.051 (±0.013) | −0.084 (±0.034) | 3.44 (±0.70) | 4.65 (±1.1) |
| LM538 | 01010 | −0.19 (±0.020) | −0.20 (±0.042) | 4.13 (±0.70) | 3.46 (±1.1) |
| LM592 | 01001 | −0.083 (±0.015) | −0.10 (±0.036) | 3.16 (±0.70) | 3.83 (±1.1) |
| LM367 | 00011 | −0.20 (±0.026) | −0.28 (±0.038) | 2.06 (±0.70) | 1.77 (±1.1) |
| LM695 | 11010 | −0.24 (±0.017) | −0.19 (±0.058) | 3.85 (±. 70) | 6.61 (±1.1) |
| LM691 | 11001 | −0.073 (±0.013) | −0.094 (±0.052) | 3.85 (±. 70) | 7.00 (±1.4) |
| LM709 | 10011 | −0.24 (±0.027) | −0.274 (±0.054) | 4.54 (±. 70) | 4.94 (±1.4) |
| LM595 | 01011 | −0.51 (±0.051) | −0.294 (±0.056) | 4.54 (±. 70) | 4.52 (±1.4) |
| LM701 | 11011 | −0.42 (±0.037) | −0.284 (±0.072) | 4.83 (±. 70) | 7.69 (±1.8) |

just $2L$ measurements (of the $r_i$ and $m_i$ values of the single mutants) instead of the measurement of $2^L$ fitness values at every concentration. In practice we do not expect a complete lack of epistasis among all mutations of interest, and the dose-response curve is also an approximation obtained by fitting a curve through a finite set of fitness values known only with limited accuracy. However, the fitness rank order of genotypes, and related topographic features such as fitness peaks, are robust to a certain amount of error in fitness values (*Crona et al., 2017*), and our model may be used to construct these to a good approximation.

Lastly, we require that the dose-response curves of the wild type and a mutant intersect at most once, which implies that the equation $w(x) = rw(\frac{x}{m})$ with $r > 1$ and $m < 1$ has at most one solution. This then also implies that the curves of any genotype $\sigma$ and a proper superset of it (i.e. a genotype which contains all the mutations in $\sigma$ and some more) intersect at most once. This property holds for all functions that have been used to represent dose-response curves in the literature, such as the Hill function, the half-Gaussian or the exponential function, as well as for all concave function with negative second derivate (see Materials and methods for details).

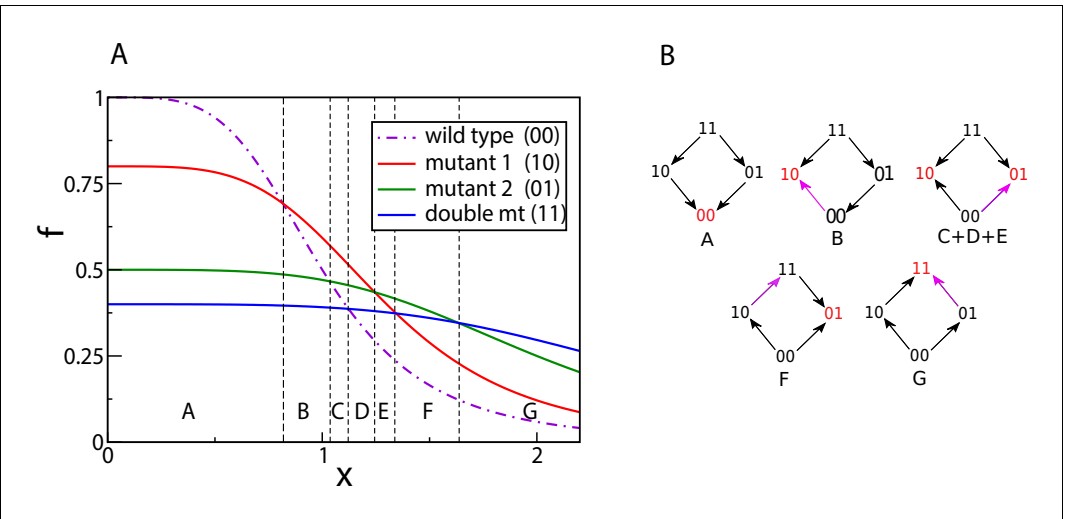

**Figure 3.** Crossing dose-response curves and fitness graphs. (**A**) An example of dose-response curves of four genotypes – the wild type (00), two single mutants (10 and 01), and the double mutant (11). The parameters of the two single mutants are $r_1 = 0.8$, $m_1 = 1.3$, $r_2 = 0.5$, $m_2 = 2.5$. Null-fitness and resistance combine multiplicatively, which implies that the parameters of the double mutant are $r_{12} = r_1 r_2 = 0.4$ and $m_{12} = m_1 m_2 = 3.25$. (**B**) Fitness graphs corresponding to antibiotic concentration ranges from panel (**A**). The genotypes in red are the local fitness peaks. The purple arrows are the ones that have changed direction at the beginning of each segment. All arrows eventually switch from the downward to the upward direction.

## Properties of tradeoff-induced fitness landscapes

To understand the evolutionary implications of our model, we first describe how the fitness landscape topography changes with the environmental parameter represented by the antibiotic concentration. Next we analyze the properties of mutational pathways leading to highly fit genotypes.

### Intersection of curves and changing landscapes

We start with a simple example of $L = 2$ mutations and a Hill-shaped dose-response curve $w(x) = \frac{1}{1+x^2}$ (*Figure 3*). At $x = 0$, the rank ordering is determined by the null-fitness. The wild type has maximal fitness, and the double mutant is less fit than the single mutants. As $x$ increases, the fitness curves start to intersect, and each intersection switches the rank of two genotypes. In the present example we find a total of six intersections and therefore seven different rank orders across the full range of $x$. This is actually the maximum number of rank orders that can be found by scanning through $x$ for $L = 2$, see Materials and methods. The final fitness rank order (in the region G in *Figure 3A*) is the reverse of the original rank order at $x = 0$. *Figure 3B* depicts the concentration-dependent fitness landscape of the 2-locus system in the form of fitness graphs. A fitness graph represents a fitness landscape as a directed graph, where neighboring nodes are genotypes that differ by one mutation, and arrows point toward the genotypes with higher fitness (*de Visser et al., 2009*; *Crona et al., 2013*).

A fitness graph does not uniquely specify the rank order in the landscape (*Crona et al., 2017*). For example, the three regions C, D and E have different rank orders but the same fitness graph. Because selection drives an evolving population towards higher fitness, a fitness graph can be viewed as a roadmap of possible evolutionary trajectories. In particular, a fitness peak (marked in red in *Figure 3B*) is identified from the fitness graph as a node with only incoming arrows. Fitness graphs also contain the complete information about the occurrences of sign epistasis. Sign epistasis with respect to a certain mutation occurs when the mutation is beneficial in some backgrounds but deleterious in others (*Weinreich et al., 2005*; *Poelwijk et al., 2007*). It is easy to read off sign epistasis for a mutation from the fact that parallel arrows (i.e. arrows corresponding to the gain or loss of the same mutation) in a fitness graph point in opposite directions.

For example, in the graph for the region B there is sign epistasis in the first position, since the parallel arrows $00 \rightarrow 10$ and $01 \leftarrow 11$ point in opposite directions. Notice that in the current

example, we start with a smooth landscape at $x = 0$ (as seen in the fitness graph for region A), and the number of peaks and the degree of sign epistasis both reach a maximum in the intermediate region C+D+E. This fitness graph displays reciprocal sign epistasis, which is a necessary condition for the existence of multiple fitness peaks (*Poelwijk et al., 2011*). Beyond the region E, the landscape starts to become smooth again, with only one fitness maximum and a lower degree of sign epistasis. In the last region G, the landscape is smooth with only one peak (the double mutant 11) and no sign epistasis.

These qualitative properties generalize to larger landscapes. To show this, we consider a statistical ensemble of landscapes with $L$ mutations, where the parameters $r_i$, $m_i$ of single mutations are independently and identically distributed according to a joint probability density $P(r, m)$. *Figure 4* shows the result of numerical simulations of these landscapes for $L = 16$. The mean number of fitness peaks with $n$ mutations reaches a maximum at $x_{\max}(n)$ where to leading order $\log x_{\max}(n) \sim n \langle \log m \rangle$, independent of any further details of the system, as argued in Materials and methods. The asymptotic expression works well already for $L = 16$ (see inset of *Figure 4A*). *Figure 4B* shows the mean number of mutations in a fitness peak. This is well approximated by the curve $n = \frac{\log x}{\langle \log m \rangle}$, showing that the mean number of mutations in a fitness peak grows logarithmically with the concentration. This is consistent with what we would expect from the variation in the number of peaks with $n$ mutations as shown in *Figure 4A*. The existence of a typical number of mutations in a fitness peak is one of the distinctive features of our landscape, a feature typically lacking in other well-studied random landscape models. This property arises from the existence of adaptational tradeoffs. Since a high number of mutations is beneficial at higher concentrations but deleterious at lower concentrations, it is clear that there must be an optimal number of mutations at some intermediate concentration.

As another indicator of ruggedness, we consider the number of backgrounds in which a mutation is beneficial as a function of $x$. At $x = 0$, any mutation is deleterious in all backgrounds, whereas at very large $x$ it is beneficial in all backgrounds. Therefore there is no sign epistasis in either case. Sign epistasis is maximized when a mutation is beneficial in exactly 1/2 of all backgrounds. *Figure 5* shows the mean number of backgrounds $n_b$ (with $n$ mutations each) in which the occurrence of a mutation is beneficial, for two different values of $n$. The curves have a sigmoidal shape, starting from zero and saturating at $\binom{L}{n}$, which is the total number of backgrounds with $n$ mutations. The blue

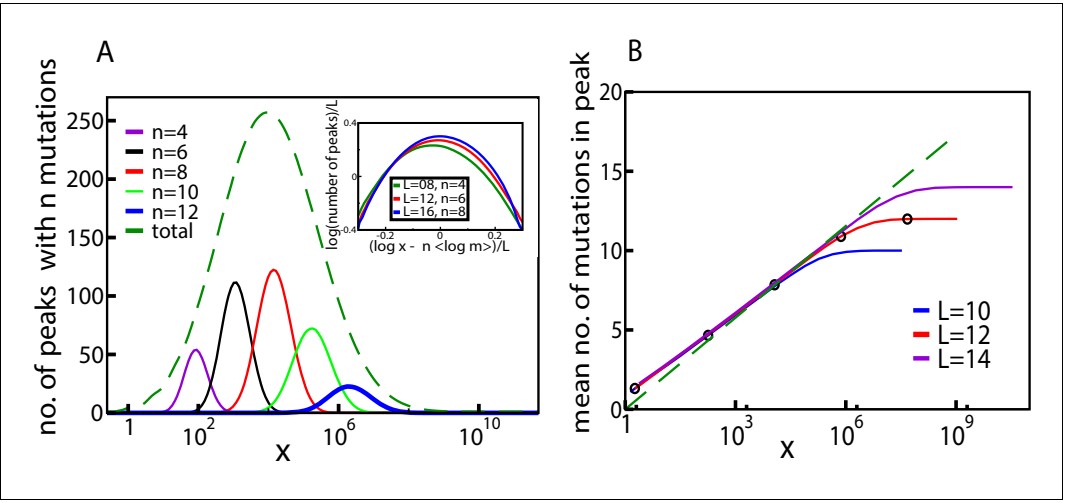

**Figure 4.** Fitness landscape ruggedness changes with drug concentration. (**A**) Number of fitness peaks as a function of concentration for different numbers of mutations in the peak, $n$, and $L = 16$. The dashed green curve is the total number of fitness peaks, summed over $n$. The peaks were found by numerically generating an ensemble of landscapes with individual effects distributed according to the joint distribution (8). For this distribution, $\langle \log m \rangle = 1.19645$. Inset: The maximal number of peaks for a given value of $n$ occurs at $\log x_{\max}(n) = n \langle \log m \rangle$, and grows exponentially with $L$. (**B**) Mean number of mutations in a fitness peak as a function of concentration $x$ for the same model. The black circles are the mean number of mutations in the fittest genotype. The green dashed line is $\frac{\log(x)}{\langle \log m \rangle}$, where $\langle \log m \rangle = 1.19645$ as before.

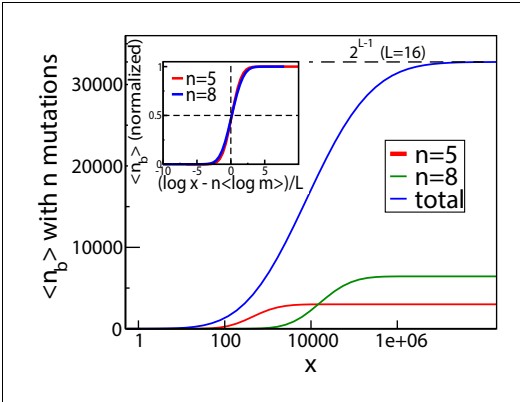

**Figure 5.** The mean number of genetic backgrounds $n_b$ in which a mutation is beneficial depends on the concentration. The numerically computed mean number is shown in the blue curve. We also computed the mean $n_b$ for genetic backgrounds with a fixed number $n$ of mutations. The results for two of these values, $n = 5$ and $n = 8$ are also shown. The inset shows these values of mean $n_b$ as a fraction of the total number of backgrounds with $n$ mutations.

curve shows the mean total number of backgrounds (with any $n$) in which a mutation is beneficial, which has a similar shape. Since every mutation in every background goes from being initially deleterious to eventually beneficial, there must be some $x$ at which every mutation is beneficial in exactly half the backgrounds. The inset of *Figure 5* shows that for backgrounds with $n$ mutations, the average concentration at which a mutation is beneficial in 1/2 the backgrounds is given by $\log x \simeq n\langle\log m\rangle$, which is the same concentration at which the largest number of fitness peaks were found in *Figure 4*. A derivation of this relation is given in Materials and methods. Similarly, when summed over all mutation numbers $n$, the fraction of beneficial backgrounds reaches 1/2 around the same concentration at which the total number of fitness peaks is maximal. Since the number of backgrounds is largest at $n = L/2$ for combinatorial reasons, this concentration is approximately given by $\log x \simeq \frac{L}{2}\langle\log m\rangle$.

Complementary to these results about the background dependence of the sign of mutational effects, it can be shown that any two distinct sets of mutations occurring in any genetic background show sign epistasis at some value of $x$. This is a consequence of the rank ordering properties of the landscapes that are described in the next subsection (see Materials and methods for a proof). A special case is that any two single mutations occurring in the wild type background must exhibit pairwise sign epistasis at some concentration.

## Accessibility of fitness peaks

Having shown that tradeoff-induced fitness landscapes display a large number of fitness peaks at intermediate concentrations, we now ask how these peaks affect the evolutionary dynamics. We base the discussion on the concept of evolutionary accessibility, which effectively assumes a regime of weak mutation and strong selection (*Gillespie, 1984*). In this regime the evolutionary trajectory consists of a series of fixation events of beneficial single-step mutations represented by a directed path in the fitness graph of the landscape (*Weinreich et al., 2005*; *Franke et al., 2011*). We say that a genotype is *accessible* from another genotype if a directed path exists from the initial to the final genotype.

The accessibility of peaks in a fitness landscape is determined by the rank ordering of the genotypes. We now show that the rank orders of tradeoff-induced fitness landscapes are constrained in a way that gives rise to unusually high accessibility. Consider two distinct sets of one or more mutations $A_i$ and $A_j$ that can occur on the genetic background $W$, and the four genotypes $W$, $WA_i$, $WA_j$ and $WA_iA_j$, where a concatenation of symbols represents the genotype which contains all the mutations referred to by the symbols. The **ordering condition** (derived in Materials and methods) says that whenever $W$ is the fittest among these four genotypes, $WA_iA_j$ must be the least fit, and whenever $WA_iA_j$ is the fittest, $W$ must be the least fit. For the case of two single mutations this situation is illustrated by the fitness graphs in *Figure 3B*, where the background genotype $W = 00$ is the fittest in the first segment A and the genotype $WA_iA_j = 11$ is the fittest in the last segment G. The ordering condition has the immediate consequence that for all environments $x$, the fittest genotype is *always* accessible from the background genotype $W$. If the fittest genotype is one of the single mutants (segments B, C, D and F), then it is of course accessible. If it is the double mutant $WA_iA_j$ (segment G), then the background genotype must be the least fit genotype (from the ordering condition), and therefore $WA_i$ and $WA_j$ should be fitter than $W$. Then $WA_iA_j$ is accessible from the wild type through the path $W \rightarrow WA_i \rightarrow WA_iA_j$ and the path $W \rightarrow WA_j \rightarrow WA_iA_j$.

To fully exploit the consequences of the ordering property we need to introduce some notation. Let $\sigma$ be a genotype with $n$ mutations. We define a *subset* of $\sigma$ as a genotype with $l$ mutations, $l \leq n$, which are all contained in $\sigma$ as well. Likewise, a *superset* of $\sigma$ is a genotype with $l$ mutations, $l \geq n$, that contains all the mutations in $\sigma$. With this, the ordering condition can be seen to imply that the superset of a fitness peak is accessible from its own supersets. For example, if $W$ is the fittest genotype, then $WA_i$ is a superset of it, and because of the ordering condition, $WA_i$ must be fitter than its superset $WA_iA_j$, and therefore accessible from it. Similarly, it is easy to show that the subset of a fitness peak is accessible from its own subsets. This property can be generalized and constitutes our main result on accessibility of fitness peaks.

**Accessibility property:** Any genotype $\Sigma$ that is a superset of a local fitness peak $\sigma$ is accessible from all the superset genotypes of $\Sigma$. Similarly, any genotype $\Sigma'$ that is a subset of a local fitness peak $\sigma$ is accessible from all the subset genotypes of $\Sigma'$.

The proof is given in Materials and methods. Three particularly important consequences are:

- Any fitness peak is accessible from all its subset and superset genotypes.
- **Any fitness peak is accessible from the wild type**. This is because the wild type is a subset of every genotype.
- For the same reason, when the wild type is a fitness peak (e.g., at $x = 0$), it is accessible from every genotype, and is therefore also the only fitness peak in the landscape. The same holds for the all-mutant when $x$ is sufficiently large, since it is a superset of every genotype.

These properties are illustrated by the fitness graph in *Figure 6*. We assume for some environment $x$ that the landscape has (at least) two peaks at the genotypes 1001 (marked in red) and 0111 (marked in blue). The colored arrows point towards mutational neighbors with higher fitness and are enforced by the accessibility property.

The edges without arrowheads are not constrained by the accessibility property and the corresponding arrows (which are not shown in the figure) could point in either direction. Consider the genotype 0111 (marked in blue). It is accessible from all its subsets, namely 0000, 0010, 0010, 0001, 0110, 0101 and 0011, following the upward pointing blue arrows. These subsets are in turn accessible from their subsets. For example, 0011 is accessible from all its subsets – 0000, 0010, and 0001. The fitness peak is also accessible from its superset 1111. The same property holds for the other fitness peak. The subsets or supersets may access the fitness peaks using other (unmarked) paths as well, which would include one or more of the undirected lines in conjunction with some of the arrows. Moreover, other genotypes, which are neither supersets nor subsets, may also access these fitness peaks through paths that incorporate some of the undirected edges.

A fitness peak together with its subset and superset genotypes defines a sub-landscape with remarkable properties. It is a smooth landscape with only one peak which is accessible from any genotype via all direct paths, that is paths where the number of mutations monotonically increases or decreases. For example, the fitness peak 1001 is accessible from the all-mutant 1111 by the two direct paths – 1111→1101→1001 and 1111→1011→1001. Likewise, the peak 0111 is accessible from its subset 0001 via the paths 0001→0101→0111 and 0001→0011→0111. In general, a peak with $n$ mutations is accessible from a subset genotype with $m$ mutations by $(n - m)!$ direct paths, and from a superset genotype with $m$ mutations by $(m - n)!$ direct paths. This gives a lower bound on the total number of paths by which a fitness peak is accessible from a subset or superset genotype.

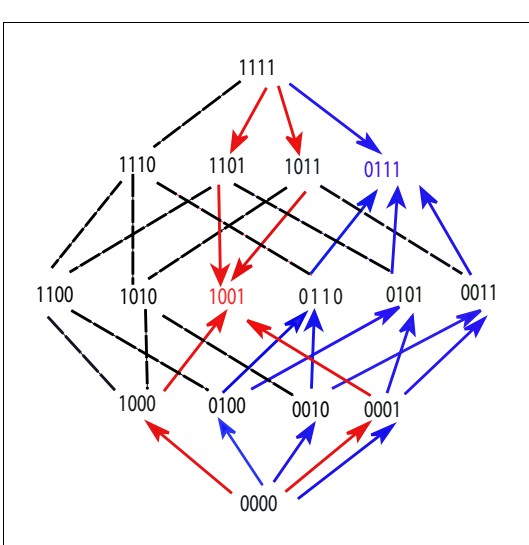

**Figure 6.** A fitness graph of a landscape with $L = 4$ mutations, illustrating the accessibility property. There are two fitness peaks, 1001 (red) and 0111 (blue). The fitness peaks are accessible from all their subset and superset genotypes following the paths marked by the arrows.

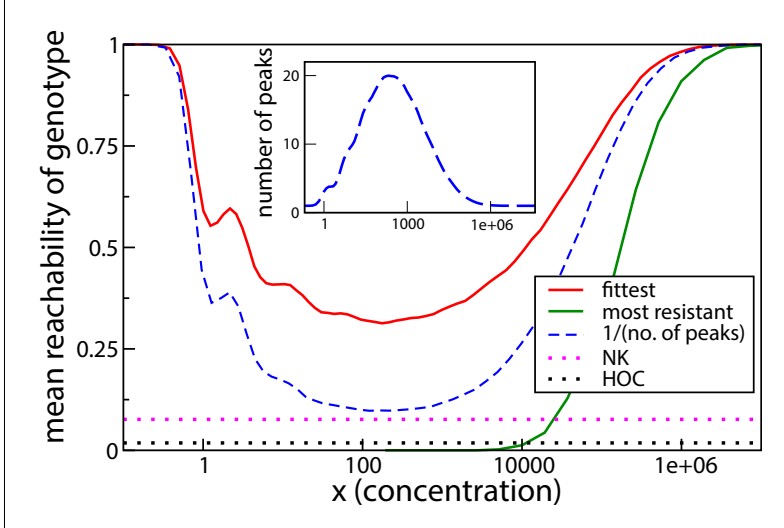

**Figure 7.** Reachability of fittest genotype and most resistant genotype. The same model as in the previous subsection has been used, with $L = 10$. Inset shows the mean number of fitness peaks as a function of concentration. Dotted horizontal lines show comparisons to the HoC model and an NK model with the same number of mutations. These models were implemented using an exponential distribution of fitness values.

Importantly, the accessibility property formulated above holds under more general conditions than stipulated in the model. We show in Materials and methods that it holds whenever the null fitness and resistance values of the mutations, $r$ and $m$, do not show *positive* epistasis. This is a weaker requirement than our original assumption of a strict lack of epistasis in these two phenotypes.

In this context it should be noted that the rank orderings forbidden by the ordering condition all show positive epistasis for the fitness values, whereas all the allowed orderings can be constructed without positive epistasis. Therefore, any landscape where positive epistasis in fitness is absent will also display the accessibility property. However, whereas the lack of positive epistasis is a sufficient condition, it is not necessary. In particular, our model does allow for cases of positive epistasis in the fitness values.

## Reachability of the fittest and the most resistant genotype

The preceding analyses have shown that within the mutant selection window, where mutants with higher fitness than the wild type exist, every fitness peak is accessible from the wild type. This includes in particular the fittest genotype at a given concentration. However, in general there will be many peaks in the fitness landscape, and it is not guaranteed that evolution will reach the fittest genotype. One can ask for the probability that the fittest genotype is actually accessed under the evolutionary dynamics, which we call its reachability. We assume that the dynamics is in the strong selection weak mutation (SSWM) regime, and the population is large enough such that the fixation probability of a mutant with selection coefficient $s$ is $1 - e^{-2s}$ for $s > 0$, and 0 for $s \leq 0$ (*Gillespie, 1984*). In our setting the selection coefficient is $s = \frac{f_1}{f_0} - 1$, where $f_1$ is the growth rate of a mutant appearing in a population of cells with growth rate $f_0$.

*Figure 7* shows the numerically obtained reachability for $L = 10$, averaged over the distribution $P(r, m)$ given in *Equation (8)*. The reachability of the highest peak is 1 at very low and very high concentrations, since there is only peak, the wild type or the all-mutant, at these extremes. The reachability is lower at intermediate concentrations, where there are multiple peaks, all of which are accessible from the wild type. The dashed blue line is the mean of the reciprocal of the total number of fitness peaks, and is therefore the mean reachability of fitness peaks. The reachability of the highest peak follows the qualitative behavior of the mean reachability, but remains higher than the mean reachability everywhere. The green curve is the reachability of the most resistant genotype, that is the all-mutant. It is extremely low at low and moderate concentrations and grows steeply and saturates quickly at a very large concentration. The all-mutant genotype is less-than-average reachable everywhere except at very high concentration, when it is the only fitness peak and accessible from every other genotype.

We have compared the reachability to that of two other widely studied landscape models. One is the House-of-Cards (HoC) model (*Kauffman and Levin, 1987*; *Kingman, 1978*), where each genotype is independently assigned a fitness value drawn from a continuous distribution. The reachability is found to be around 0.018, an order of magnitude smaller than the lowest reachability seen in the tradeoff-induced landscape. The mean number of fitness maxima in the HoC landscape is $\frac{2^L}{L+1}$, which in this case is approximately 93.1, much higher than the maximum mean number of peaks in the tradeoff-induced landscape (inset of *Figure 7*). We would therefore naturally expect a smaller fraction of adaptive walks to terminate at the fittest peak. A more illuminating comparison is with the NK model (*Kauffman and Weinberger, 1989*; *Hwang et al., 2018*). Here, once again, $L = 10$, and

the mutations are divided into two blocks of 5 mutations each. As per the usual definition of the model, the fitness of a genotype is the sum over the contributions of each of the 10 mutations, and the contribution of each mutation depends only the state of the block to which it belongs. The fitness contribution of each mutation for any state of the block is an independent random number. The mean number of fitness maxima here is $\simeq 28.44$ (*Perelson and Macken, 1995*; *Schmiegelt and Krug, 2014*), which is comparable to the maximum mean number in the tradeoff-induced landscapes (see inset of *Figure 7*). Nonetheless, the reachability of the fittest peak (dotted pink line) is found to be nearly 4 times smaller than the lowest reachability in our landscape. We found that in a fraction of about 0.64 of the landscapes, the fittest maximum is not reached in any of 32000 dynamical runs, indicating the absence of an accessible path in most of these cases (*Schmiegelt and Krug, 2014*; *Hwang et al., 2018*). In contrast, an evolutionary path always exists to any fitness peak in the tradeoff-induced landscapes, as we saw in the previous subsection. This endows the tradeoff-induced landscapes with the unusual property of being highly rugged and at the same time having a much higher evolutionary reachability of the global fitness maximum compared to other models with similar ruggedness.

## Discussion

Fitness landscapes depend on the environment, and gene-gene-interactions can be modified by the environment. Systematic studies of such $G \times G \times E$ interactions are rare, but they are clearly of relevance to scenarios such as the evolution of antibiotic resistance, where the antibiotic concentration can vary substantially in space and time. In this paper we have explored the structure of such landscapes in the presence of tradeoffs between fitness and resistance. We summarize the main findings of our work.

- We have shown experimental evidence that the dose-response curves of various mutant strains of *E. coli* to the antibiotic ciprofloxacin have the same shape, except for a rescaling of the fitness and concentration values. If this shape is known, the fitness of a strain can be estimated at any antibiotic concentration simply by measuring its null-fitness and $IC_{50}$ (or MIC). This makes it possible to construct empirical fitness landscapes at any antibiotic concentration from a limited set of data.
- Under the assumptions of our model the degree of epistasis, particularly sign epistasis, is low for zero and high antibiotic concentrations, but it is nevertheless high in the intermediate concentration regime. The number of local fitness peaks scales exponentially in the number of mutations at these concentrations. Epistasis is often discussed as a property intrinsic to mutations and their genetic backgrounds, with limited consideration of environmental parameters. But in the landscapes studied here, the environmental parameter is of paramount importance, since changes in it can dramatically alter gene-gene interactions.
- The expected number of mutations in a fitness peak increases logarithmically with the antibiotic concentration. This implies that, at a given concentration, the highly fit genotypes that make up the fitness peaks carry an optimal number of mutations that arises from the tradeoff between fitness cost and resistance.
- Despite the high ruggedness, the landscape displays strong non-random patterns. A rank ordering condition between sets of mutations holds at all concentrations. A remarkable and unexpected consequence of this is that any fitness peak is evolutionarily accessible from the wild type.
- It is well known from experimental studies of antimicrobial resistance evolution that highly resistant genotypes often require multiple mutations which can be acquired along different evolutionary trajectories. Epistatic interactions constrain these trajectories and are generally expected to impede the evolution of high resistance. We find that strong and complex epistatic interactions inevitably arise in the mutant selection window, but at the same time the evolution of the most resistant genotype (the identity of which changes with concentration) remains facile and can occur along many different pathways.

All of these conclusions follow from three basic assumptions that are readily generalizable beyond the context of antimicrobial resistance evolution: the existence of tradeoffs between two *marginal phenotypes* that govern the adaptation at extreme values of an environmental parameter; the scaling property of the shape of the tradeoff function; and the condition of limited epistasis for the marginal phenotypes. How generally these assumptions are valid is a matter of empirical investigation.

We have shown that they hold for certain cases, and the interesting evolutionary implications of our results indicate that more empirical research in this direction will be useful.

In the case of antimicrobial resistance, there can be fitness compensatory mutations (*Levin et al., 2000*; *Brown et al., 2010*; *Durão et al., 2018*) that do not exhibit any adaptational tradeoffs. These mutations are generally found in a population in the later stages of the evolution of antibiotic resistance, which implies that they emerge in a genetic background of mutations with adaptational tradeoffs. An understanding of tradeoff-induced landscapes is therefore a prerequisite for predicting the emergence of compensatory mutations. While compensatory mutations are expected to facilitate the evolution of high resistance (*Hughes and Andersson, 2017*), our study shows that the acquisition of multiple resistance mutations may readily occur even if compensatory mutations are absent.

In the formulation of our model we have assumed for convenience that the marginal phenotypes combine multiplicatively, but this assumption is in fact not necessary for all our results. As shown in Materials and methods, our key results on accessibility only require the absence of positive epistasis. These results therefore hold without exception for the combinatorially complete data set in *Table 1*, where epistasis is either absent or negative. More generally, our analysis remains valid in the presence of the commonly observed pattern of diminishing returns epistasis among beneficial mutations (*Chou et al., 2011*; *Schoustra et al., 2016*; *Wünsche et al., 2017*). We expect our results to hold approximately even when there is a small degree of epistasis (positive or negative) in $r$ and $m$, but we do not explore that question quantitatively in this paper.

A strict absence of epistasis, while certainly not universal, can be expected to occur under certain generic circumstances. Assuming that we deal with a single antibiotic that has a single target enzyme, we can think of two situations that could lead to a multiplicative behavior of the $IC_{50}$: (i) Single mutations occur in different genes that affect the concentration of the antibiotic-target enzyme complex through independent mechanisms. (ii) Single mutations occur in the same gene but their effect is multiplicative due to the nature of antibiotic-enzyme molecular interactions. An example of scenario (i) would be a combination of mutations in the target gene (reduction of the binding affinity), its promoter (increase in expression), genes regulating the activity of efflux pumps and porins (decrease in intracellular concentration of the antibiotic), or genes controlling the level (increase in concentration) or activity of drug-degrading enzymes. These mechanisms are 'orthogonal' to each other, in the sense that they modify independent pathways within the cell. If each of them affects the concentration of the antibiotic-target complex through first-order kinetics, their cumulative effect will be multiplicative in terms of the $IC_{50}$s of single mutations.

In the case of ciprofloxacin and *E. coli* (*Figure 2* and *Table 1*), we expect mutations in *gyrA* (target) to be orthogonal to mutations in *acrR* and *marR* (efflux pumps). This is borne out by the observed multiplicativity of the $IC_{50}$ (*Table 1*). In contrast, we expect scenario (ii) to apply if the single mutations affect different parts of the antibiotic-enzyme binding site independently. This is not the case for two particular mutations in *gyrA* studied here – S83L and D87N (see cases of epistasis in *Table 1*). An example for scenario (ii) are the two mutations P21L and A26T in the gene encoding the enzyme dihydrofolate reductase, which increase the resistance to trimethoprim in a multiplicative way in the absence of other mutations (*Palmer et al., 2015*). If the antibiotic has more than one target, multiplicativity would not generally hold. In particular, topoisomerase IV (gene *parC*) is a secondary target for ciprofloxacin with much weaker affinity than gyrase. Therefore, mutations in *parC* do not contribute to resistance unless there is already a mutation in *gyrA*. As a consequence, in contrast to the mutants listed in *Table 1*, combinations containing parC display positive epistasis (*Hughes and Andersson, 2017*).

The co-existence of high ruggedness and high accessibility found in the tradeoff-induced landscapes studied here is counterintuitive, and to the best of our knowledge fitness landscape models with this property have not been described previously. The situation is depicted schematically in *Figure 8*. The first landscape is smooth with a single peak that must be accessible from everywhere else. The second landscape is rugged, and each fitness peak is typically accessible from a few genotypes only. This is the typical picture of a rugged fitness landscape with limited accessibility, as it would be predicted by simple statistical models such as the HoC, NK or rough Mt. Fuji models (*Szendro et al., 2013*; *Neidhart et al., 2014*; *Hwang et al., 2018*). The landscapes we describe here belong to a third type, where a high number of peaks are accessible from a high number of genotypes, creating overlapping 'valleys' from which a population may evolve towards different local fitness maxima. Moreover, not only are fitness peaks accessible from all their subset and superset

**Figure 8.** Accessibility and ruggedness in different types of fitness landscapes. The first two landscapes correspond to the typical cases of smooth and rugged landscapes. The third figure describes landscapes with adaptational tradeoffs, where high ruggedness coexists with high accessibility.

genotypes, but there are many direct paths leading up to each peak. This appears contrary to the expectation that in landscapes with high epistasis, accessibility should be facilitated through mutational reversions, that is, indirect paths (*DePristo et al., 2007*; *Palmer et al., 2015*; *Wu et al., 2016*; *Zagorski et al., 2016*).

We conclude with some possible directions for future work. Our model provides a principled framework for predicting how microbial fitness landscapes vary across different antibiotic concentrations. This could be exploited to describe situations where the antibiotic concentration varies on a time scale comparable to the evolution of resistance, either due to the degradation of the drug or by an externally imposed treatment protocol (*Marrec and Bitbol, 2018*). In this context it would be of particular interest to include compensatory mutations that lack the tradeoff between growth and resistance, since such mutations are expected to strongly affect the extent to which resistance can be reversed (*Andersson and Hughes, 2010*). Significant extension of the theory is required if the drug concentration varies on a faster time scale comparable to the growth time of the microbial population, in which case the concept of a concentration-dependent fitness would need to be reconsidered.

From the broader perspective of evolutionary systems with adaptational tradeoffs mediated by an environmental parameter, our study makes the important conceptual point that it is impossible to have non-epistatic fitness landscapes for all environments. Using the terminology of *Gorter et al., 2016*, the tradeoffs enforce reranking $G \times E$ interactions which in turn, as we have shown, induce sign-epistatic $G \times G$ interactions at intermediate values of the environmental parameter. Notably, this general conclusion does not depend on the scaling property of the tradeoff function. It would nevertheless be of great interest to identify instances of scaling for other types of adaptational tradeoffs, in which case the detailed predictions of our model could be applied as well.

# Materials and methods

## Experiments
### Bacterial strains
We used strains from *Marcusson et al., 2009* (courtesy of Douglas Huseby and Diarmaid Hughes). The strains are isogenic derivatives of MG1655, a K12 strain of the bacterium *E. coli*, with specific point mutations or gene deletions in five different loci: *gyrA:S83L*, *gyrA:D87N*, *parC:S80I*, Δ*marR*, and Δ*acrR*. There are 32 possible combinations of these alleles, but we only used the wild type, single mutants (five strains) and double mutants (8 strains of 10 possible combinations): LM179 (00000), LM378 (10000), LM534 (01000), LM792 (00100), LM202 (00010), LM351 (00001), LM625 (11000), LM862 (10100), LM421 (10010), LM647 (10001), LM1124 (01100), LM538 (01010), LM592 (01001), LM367 (00011). A binary sequence after the strain's name represents the presence/absence of a particular mutated allele (order as in the above list of genetic alterations).

### Growth media and antibiotics

LB growth medium was prepared according to Miller's formulation (10 g tryptone, 5 g yeast extract, 10 g NaCl per litre). The pH was adjusted to 7.2 with NaOH, and autoclaved at 121°C for 20 min. Ciprofloxacin (CIP) solutions were prepared from a frozen stock (10 mg/ml ciprofloxacin hydrochloride, pharmaceutical grade, AppliChem, Darmstadt, in sterile, ultra-pure water) by diluting into LB to achieve the desired concentrations.

### Dose-response curves

We incubated bacteria in 96-well clear flat bottom micro-plates (Corning Costar) inside a plate reader (BMG LABTECH FLUOstar Optima with a stacker) starting from two different initial cell densities (half a plate for each), and measured the optical density (OD) of each culture every 2–5 min to obtain growth curves. Plates were prepared automatically using a BMG LABTECH CLARIOstar plate reader equipped with two injectors connected to a bottle containing LB and a bottle with a solution of CIP in LB. The injectors were programmed to create different concentrations of CIP in each column of the 96 well plate. The injected volumes of the CIP solution were 0, 20, 25, 31, 39, 49, 62, 78, 98, 124, 155, 195 μl, and an appropriate volume of LB was added to bring the total volume to 195 μl per well. Since different strains had MICs spanning almost two decades of CIP concentrations, we used a different maximum concentration of the CIP solution for each strain (approximately 1.5–2 times the expected MIC). Bacteria were diluted from a thawed frozen stock $10^3$ and $10^4$ times in PBS (phosphate buffered saline buffer), and 5μl of the suspension was added to each well ($10^3$ dilution to rows A-D, $10^4$ dilution to rows E-H). We used one strain per plate and up to four plates per strain (typically 1–2). After adding the suspension of bacteria to each well, the plates were immediately sealed with a transparent film to prevent evaporation, and put into a stacker (37°C, no shaking), from which they would be periodically fed into the FLUOstar Optima plate reader (37°C, orbital shaking at 200 rpm for 10 s prior to OD measurement). We then used the time shift method to obtain exponential growth rates for each strain and different concentrations of CIP, see *Ojkic et al., 2019* for further details.

## Mathematical methods

### Rank orders and fitness graphs

The total number of possible rank orders with $L$ mutations is $2^L!$, which is 24 for $L = 2$. Not all these rank orders, however, can be realized as one scans through $x$. Since any two curves intersect at most once, the maximum number of distinct rank orders that can be reached is the rank order at $x = 0$ plus the total number of possible intersections, which is $\binom{2^L}{2} = 2^{L-1}(2^L - 1)$. Thus the upper bound on the number of rank orders found by scanning through $x$ is $2^{L-1}(2^L - 1) + 1$, which is smaller than $2^L!$ for $L \geq 2$.

It is also instructive to determine the number of fitness graphs that can be found by varying $x$ for a system with $L$ mutations. This can be computed as follows: At $x = 0$ every mutation is deleterious, and every mutational neighbor with one less mutation is fitter; but due to the tradeoff condition, at sufficiently large $x$ every mutation is beneficial and any mutational neighbor with one less mutation is less fit. In order for this reversal of fitness order to happen, the dose-response curves of any two mutational neighbors must intersect at some $x$. Therefore, the number of fitness graphs generated is equal to the number of distinct pairs of mutational neighbors, which is $2^{L-1}L$, and the number of distinct fitness graphs encountered is $2^{L-1}L + 1$. For $L = 2$, this number is 5, as seen in the example in the main text.

### Condition for two dose-response curves to intersect at most once

Consider two DR curves characterized by $(r, m)$ and $(r', m')$, where $r < r'$ and $m > m'$. We need to show that for the commonly observed cases, the curves $rw(\frac{x}{m})$ and $r'w(\frac{x}{m'})$ intersect at most once. First, notice that it is sufficient to prove this for the case $r' = 1, m' = 1$, because any rescaling of the $x$ and $w$ axes does not alter the number or ordering of intersection points. Therefore we require $r < 1$ and $m > 1$.

Let us consider the case where the dose-response curve is of the form of a Hill function, that is $w(x) = \frac{1}{1+x^a}$, with $a>0$. The intersection of curves happens at the solution of $w(x) = rw(\frac{x}{m})$, which we denote by $x^*(r,m)$. In this case the solution is given by

$$x^*(r,m) = \left(\frac{1-r}{r-\frac{1}{m^a}}\right)^{\frac{1}{a}}$$

which is positive and unique if $rm^a>1$; otherwise no solution with $x^*>0$ exists. It is similarly easy to show that at most one intersection point exists for exponentials, stretched exponentials, and half-Gaussians.

The property also holds for any concave dose-response curve with $w''(x)<0$. We prove this as follows. Any intersection point $x^*$ is the solution of

$$F(x^*) = r$$

where $F(x) \equiv \frac{w(x)}{w(\frac{x}{m})}$. We will show that $F(x)$ is monotonic and therefore the above equation has at most one solution. We have

$$F'(x) = \frac{w'(x)w(\frac{x}{m}) - \frac{1}{m}w(x)w'(\frac{x}{m})}{w(\frac{x}{m})^2},$$

and $F'(x)$ has the same sign as the numerator $\mathcal{N}(x) = w'(x)w(\frac{x}{m}) - \frac{1}{m}w(x)w'(\frac{x}{m})$. Since $w(x)$ is a decreasing function and $m>1$, $w(\frac{x}{m})>w(x)>\frac{1}{m}w(x)$. When $w''(x)<0$, we also have $w'(x)<w'(\frac{x}{m})$. Since $w'(x)<0$, this implies $|w'(x)|>|w'(\frac{x}{m})|$, and $\mathcal{N}(x)<0$. Therefore $F(x)$ is monotonically decreasing.

## Proof of the accessibility property

To derive the ordering condition, let us start with the simplest case of two single mutations $A_i$, $A_j$ occurring on the wild type background. There are correspondingly four different genotypes $W$, $WA_i$, $WA_j$, $WA_iA_j$, which are listed in decreasing order of fitness at $x=0$. Let the intersection of the DR curves of two genotypes $\sigma_1$ and $\sigma_2$ occur at $x = X_{\sigma_1,\sigma_2}$. Then $X_{W,WA_j}$ is given by the solution $x^*(r_j, m_j)$ of

$$w(x) = r_j w(\frac{x}{m_j}),$$

and $X_{WA_i,WA_iA_j}$ is given by the solution of

$$r_i w(\frac{x}{m_i}) = r_i r_j w(\frac{x}{m_i m_j}).$$

This last equation can be re-written as

$$w(x') = r_j w(\frac{x'}{m_j}),$$

where $x' = \frac{x}{m_i}$. Comparing this with the first equation above, we have

$$X_{WA_i,WA_iA_j} = m_i X_{W,WA_j} > X_{W,WA_j}. \tag{3}$$

This equation tells us that whenever the double mutant is fitter than one of the single mutants, the wild type must be less fit than the *other* single mutant. Consequently, when the double mutant is fitter than both the single mutants, the WT must be less fit than both the single mutants. In other words, the number of single mutants fitter than the wild type cannot be less than the number of single mutants less fit than the double mutant. This is the ordering condition given in the main text. Any ordering that violates this condition is a *forbidden ordering*. For greater clarity, we list all the possible forbidden orderings (up to interchange of indices *i* and *j*).

$$W > WA_i > WA_iA_j > WA_j$$
$$W > WA_iA_j > WA_i > WA_j$$
$$WA_iA_j > W > WA_i > WA_j$$
$$WA_iA_j > WA_i > W > WA_j$$

(4)

Although we showed this for two single mutations in the wild type background, the same arguments hold for any two sets of mutations in any background, since the succession of orderings is independent of the rescalings of the fitness and concentration axes. To put it more precisely, $W$, $A_i$ and $A_j$ are any three non-overlapping sets of mutations, where $A_i$ and $A_j$ are non-empty sets.

Next we use this to prove the accessibility property. Let $\sigma$ have $n$ mutations. It is sufficient to prove that (i) any superset of $\sigma$ with $m$ or fewer mutations is accessible from all its own supersets with $m$ or fewer mutations, for all $m \geq n$ (the statement follows from the case $m = L$); and that (ii) any subset of $\sigma$ with $m'$ or more mutations is accessible from any of its own subsets with $m'$ or more mutations, for all $m' \leq n$ (the statement corresponds to $m' = 0$). We prove this by induction.

Firstly, we notice that the case $m = n$ is trivial, since $\sigma$ is of accessible from itself. For the case of supersets, our base case is $m = n + 1$, and the assertion above holds because $\sigma$ is a local fitness peak, and therefore accessible from all its supersets with $n + 1$ mutations, which are of course accessible from themselves.

Now we prove the induction step. Assume that all supersets of $\sigma$ that have $m$ or fewer mutations (where $m \geq n$) are accessible from all their supersets with $m$ or fewer mutations. Consider a superset $\Sigma$ of $\sigma$ with $m$ mutations, and denote it by $\Sigma = \sigma A$, where $A$ is the set of mutations in $\Sigma$ not present in $\sigma$. By assumption, $\sigma$ is accessible from $\Sigma$. In the following, we use the notation $\sigma_1 > \sigma_2$ to indicate that a genotype $\sigma_1$ is fitter than a genotype $\sigma_2$ (we use the '<' and '=' signs in a similar way). Therefore, we have $\sigma > \Sigma = \sigma A$.

Now consider any superset of $\Sigma$ with $m + 1$ mutations, where the additional mutation not contained in $\Sigma$ is denoted $B$. Then this superset can be denoted by $\Sigma B = \sigma AB$. We must have $\sigma > \sigma B$ since $\sigma$ is a local fitness peak. We now have the relation $\sigma > \sigma A, \sigma B$. Therefore we must have $\sigma AB < \sigma A, \sigma B$, for otherwise we violate the ordering condition. Now since $\Sigma B = \sigma AB < \sigma A = \Sigma$, $\Sigma$ must be accessible from $\Sigma B$, proving that any superset with $m$ mutations is accessible from any of its supersets with $m + 1$ mutations. This completes the proof of the induction step.

The proof for the case of subsets is essentially the same, utilizing the symmetry between the wild type and the double mutant in the ordering condition.

The accessibility property follows entirely from the ordering condition, and hence any landscape that obeys the ordering condition will obey the theorem. The ordering condition follows from $X_{W,WA_i} < X_{WA_j,WA_iA_j}$, as obtained in *Equation 3*. However, this same inequality obtains under more general conditions. To see this, let us define the null-fitness of the double mutant $WA_iA_j$ as $r_{ij}$, and the resistance of the double mutant as $m_{ij}$. The dose-response curves of $W$ and $WA_j$ intersect at $X_{W,WA_j} = x^*(r_j, m_j)$, whereas the curves for $WA_i$ and $WA_iA_j$ intersect at

$$X_{WA_i,WA_iA_j} = m_i x^*\left(\frac{r_{ij}}{r_i}, \frac{m_{ij}}{m_i}\right).$$

Now it is easy to show that $x^*(r, m)$ is a decreasing function of both $r$ and $m$. Therefore $X_{WA_i,WA_iA_j} > X_{W,WA_j}$ holds if $r_{ij} \leq r_i r_j$ and $m_{ij} \leq m_i m_j$.

## Number of local fitness peaks

When dealing with complex fitness landscapes with parameters that can vary across species and environments, a useful strategy is to model the fitness effects as random variables that are chosen from a probability distribution (*Kauffman and Levin, 1987*; *Szendro et al., 2013*; *Hwang et al., 2018*). In the limit of large system size $L$, many properties emerge that are independent of the details of the system. In practice, even relatively small system sizes are often approximated well by results obtained in the asymptotic limit.

The mean number of peaks with $n$ mutations in the tradeoff-induced landscapes is

$$K_n(x) = \binom{L}{n} Q_n(x),$$

where $\binom{L}{n}$ is the total number of genotypes with $n$ mutations, and $Q_n(x)$ is the probability that a genotype with $n$ mutations is a fitness maximum at antibiotic concentration $x$. Then the total number of peaks at $x$ is $\sum_n K_n(x)$. Let the resistance of a genotype $\sigma$ be $M = \prod_{i=1}^n m_i$, and likewise its null-fitness be $R = \prod_{i=1}^n r_i$. The genotype $\sigma$ is a local fitness maximum if it is fitter than all its subsets with $n-1$ mutations and all its supersets with $n+1$ mutations.

To find the concentration at which the curves of $\sigma$ and its neighboring genotypes intersect, we start with the simplest case of the dose-response curves of the wild type and a single mutant $(r, m)$. These curves intersect at the solution $x^*(r, m)$ of $w(x) = rw(\frac{x}{m})$, which is a decreasing function of $r$ and $m$. The wild type is fitter than the single mutant when $x < x^*(r, m)$. Now the intersection of the DR curves of a genotype $\sigma$ with $n$ mutations and a subset with $n-1$ mutations that lacks the mutation $(r_i, m_i)$ occurs at the solution of

$$w\left(\frac{x}{\left(\frac{M}{m_i}\right)}\right) = r_i w\left(\frac{x}{\left(\frac{M}{m_i}\right)m_i}\right)$$

which is read off as $\frac{M}{m_i}x^*(r_i, m_i)$. Likewise, the intersection of the DR curves of $\sigma$ and a superset with $n+1$ mutations that contains the additional mutation $(r_j, m_j)$ occurs at $Mx^*(r_j, m_j)$. Therefore $\sigma$ is a fitness maximum if

$$\frac{x^*(r_i, m_i)}{m_i} < \frac{x}{M} < x^*(r_j, m_j) \tag{5}$$

for all $i$ and $j$ with $1 \leq i < n$ and $n < j \leq L$. Alternatively,

$$\log m_i - \log x^*(r_i, m_i) > \log M - \log x > -\log x^*(r_j, m_j). \tag{6}$$

Let us consider the regime where $L, n \gg 1$. Then $\log M \sim n\langle \log m\rangle$; if $\log x$ is smaller than $O(n)$, it is clear that the second inequality is almost certainly satisfied whereas the probability of the first inequality is vanishingly small. Both the probabilities are finite if $\log x \sim n\langle \log m\rangle$. Thus the probability of $\sigma$ being a fitness peak is maximized when $\log x = \log(M) + \eta$, where $\eta \sim O(1)$ and depends on the details of the distribution $P(r, m)$. Thus the mean number of fitness peaks with $n$ mutations is maximal at $x_{\max}(n)$ where to leading order $\log x_{\max}(n) \sim n\langle \log m\rangle$, independent of any further details of the system.

The total number of genotypes with $n$ mutations is $\binom{L}{n}$, and $\log \binom{L}{n} \simeq LH(\rho)$, where $\rho = \frac{n}{L}$, and

$$H(\rho) = -[\rho \log \rho + (1-\rho)\log(1-\rho)]. \tag{7}$$

The mean number of fitness maxima can be found by multiplying this with $Q_n$. One may expect $Q_n$ to be exponentially small in $L$, since a total of $L$ inequalities (as indicated in *Equation 6*) need to be satisfied. However, this is complicated by the fact that the probabilities of the inequalities being satisfied are not independent. The correlations between the inequalities would depend on the distribution of $P(r, m)$ and the dose-response curve. If the correlations are sufficiently weak, one might still expect to find an exponential scaling in large $L$. To leading order $\binom{L}{n}$ is itself exponential in $L$, and if the probability that a genotype is a fitness peak is exponentially small in $L$, we expect the mean number of peaks $K_n$ to be exponential in $L$ as well. This is supported by the scaling shown in the inset of *Figure 4A*.

For the simulation results shown in the main text we chose a joint distribution of the form

$$P(r, m) = P(r)P(m|r) = 6r(1-r)(m - \frac{1}{\sqrt{r}})e^{-(m - \frac{1}{\sqrt{r}})}. \tag{8}$$

The conditional distribution $P(m|r)$ is a shifted gamma distribution. The shift ensures that the curves of a background genotype and a mutant intersect.

## Sign epistasis

Sign epistasis with respect to a certain mutation occurs when the mutation is beneficial in one background but deleterious in another. We first show that any two distinct sets of mutations on any genetic background display sign epistasis at some value of the scaled concentration $x$. Consider a

genetic background $W$, and two distinct sets of mutations $A_1$ and $A_2$ (which share no mutations with each other or $W$). At $x = 0$ we have $W > A_1$, $A_2$ and $WA_1A_2 < WA_1$, $WA_2$. As $x$ increases, $W$ must become less fit than either $WA_1$ or $WA_2$ before $WA_1A_2$ becomes fitter than either of these (by the ordering condition). Without loss of generality, let us assume that $W$ becomes less fit than $WA_1$ before it becomes less fit than $WA_2$. At this point, we must have $W < WA_1$ and $WA_2 > WA_1A_2$. This means that, in the wild type background, $A_1$ is beneficial in the absence of $A_2$ but deleterious in the presence of $A_2$, indicating pairwise sign epistasis.

To quantify the amount of sign epistasis for large $L$ and $n$, we next ask for the number of backgrounds $n_b$ in which a mutation is beneficial at concentration $x$. If one considers only those backgrounds that have $n$ mutations, then $n_b$ would depend both on $n$ and $x$. In a statistical ensemble of landscapes, one may compute the probability $P_b$ that a mutation is beneficial in a background with $n$ mutations, and of course $\langle n_b \rangle = P_b \binom{L}{n}$. In the limit of large $L$ and $n$, $P_b$ exhibits some universal properties to leading order. When $\log x > n \langle \log m \rangle$, we are in the regime of high concentration relative to $n$, and we expect a mutation to be beneficial. We find that to leading order $P_b(\rho, x) = 1$, with corrections that are exponentially small in $n$. When $\log x < n \langle \log m \rangle$, we are at concentrations that are too low to prefer additional mutations, and $P_b$ is exponentially small in $n$. When $\log x = n \langle \log m \rangle$, we are at the threshold concentration where a new mutation becomes beneficial. Here we find that $P_b \simeq \frac{1}{2}$. For large $L$ we therefore expect a steep transition from 0 to 1 as the concentration crosses the threshold value (see inset of *Figure 5*).

Consider a mutation $(r, m)$ in a background with $n$ mutations $(r_1, m_1), (r_2, m_2) \ldots (r_n, m_n)$. The mutation is beneficial in this background if

$$m_1 m_2 \ldots m_n x^*(r, m) < x \tag{9}$$

Taking logarithms, we have

$$-\log x^*(r, m) > \sum_{i=1}^{n} \log m_i - \log x. \tag{10}$$

Define $\xi = \frac{\log x}{L}$ and $\rho = \frac{n}{L}$, and $z = -\log x^*(r, m)$. Then the above inequality becomes

$$\frac{z}{n} > \frac{1}{n} \sum_{i=1}^{n} \log m_i - \frac{\xi}{\rho}. \tag{11}$$

Let the distribution of $z$ be $P(z)$, and let $C_z(z) = \int_z^\infty P_z(x) \, dx$. Define the random variable $\omega = \frac{1}{n} \sum_{i=1}^{n} (\log m_i - \frac{\xi}{\rho})$, and denote its distribution $P(\omega)$. Then the probability that a mutation is beneficial in a background with $n$ mutations is

$$P_b(\rho, \xi) = \int_{-\infty}^{\infty} P(\omega) \, C_z(n\omega) \, d\omega. \tag{12}$$

The mean number of backgrounds with $n$ mutations in which a mutation is beneficial is $n_b(\rho, \xi) = P_b(\rho, \xi) \binom{L}{n}$. Note that $\langle \omega \rangle = \langle \mu \rangle - \frac{\xi}{\rho}$ where $\mu = \log m$. When $n \gg 1$, $C_z(n\omega) \simeq 1$ for $\omega < 0$ and $C_z(n\omega) \simeq 0$ for $\omega > 0$, with a sharp transition from 1 to 0 that happens within a region of width $\sim O(1/n)$ of the origin. Also for large $n$, $P(\omega)$ is sharply peaked around $\langle \omega \rangle$ over a region of width $O(1/\sqrt{n})$.

When $\langle \omega \rangle < 0$, $C_z(n\omega) \simeq 1$ over this entire region, as observed before. Thus to leading order, $P_b(\rho, \xi) = 1$. The mean number of backgrounds in which a mutation is beneficial is $n_b(\rho, \xi) = P_b(\rho, \xi) \binom{L}{\rho L}$.

$$n_b(\rho, \xi) \simeq \sqrt{\frac{2\pi}{L}} \frac{1}{\sqrt{\rho(1-\rho)}} e^{LH(\rho)} \tag{13}$$

where $H(\rho)$ is defined in *Equation 7*. Therefore

$$\log n_b \simeq LH(\rho) \tag{14}$$

to leading order.

When $\langle\omega\rangle > 0$, the dominant contribution to the integral in (*Equation 12*) comes from $\omega \leq 0$, since $C_z(n\omega)$ quickly drops from 1 to zero for $\omega > 0$. Further, since $C_z(\omega) \simeq 1$ for $\omega < 0$ (except for a region of width $O(1/n)$ around $\omega = 0$, as observed before), we can approximate $\log P_b(\rho, \xi)$ simply by the probability that $\omega < 0$. Then

$$\log P_b(\rho, \xi) \simeq -nI\left(-\frac{\xi}{\rho}\right)$$

where $I$ is the large deviation function of $-\mu$, and

$$\log n_b(\rho, \xi) \simeq L\left[H(\rho) - \rho I\left(-\frac{\xi}{\rho}\right)\right].$$

This implies that $n_b$ is reduced by a factor that is exponentially small in $L$ compared to *Equation 14*, and therefore the fraction of backgrounds in which a mutation is beneficial is very small.

Finally, when $\langle\omega\rangle = 0$, that is $\xi = \frac{n}{L}\langle\mu\rangle$, $P(\omega)$ is centered at the origin and decays over a width $O(1/\sqrt{n})$. For $\omega > 0$, $C_z(n\omega)$ is 0 except over a much smaller width $O(1/n)$ to the right of the origin, whereas for $\omega \leq 0$, it is 1 except for a small region of width $O(1/n)$ left of the origin. Thus the dominant contribution to the integral in *Equation 12* comes from $\omega \leq 0$, and as before, $P_b$ can be approximated by the probability that $\omega \leq 0$. Due to the central limit theorem, $P(\omega)$ is approximately Gaussian and therefore symmetric around $\omega = 0$, and therefore $P_b \simeq \frac{1}{2}$. Consequently, we should have

$$n_b(\rho, \xi) \simeq \frac{1}{2}\sqrt{\frac{2\pi}{L}}\frac{1}{\sqrt{\rho(1-\rho)}}e^{LH(\rho)},$$

which is 1/2 times the total number of backgrounds given by *Equation 13*. This proves that the concentration where the mutation is beneficial in half of the backgrounds is given by $\langle\omega\rangle = 0$ or $\log x = n\langle\log m\rangle$ for large $L$ and $n$.

# Acknowledgements

We thank Douglas Huseby and Diarmaid Hughes for providing us with the *E. coli* strains of *Marcusson et al., 2009*, and Tobias Bollenbach, Michael Brockhurst and Kristina Crona for useful comments. The work of SGD and JK was supported by DFG within CRC 1310 *Predictability in Evolution*, and JK acknowledges the kind hospitality of the Scottish Universities Physics Alliance and the Higgs Center for Theoretical Physics during the completion of the project. SOLD and RJA acknowledge the support of the ERC Consolidator Grant 682237 EVOSTRUC.

# Additional information

## Funding

| Funder | Grant reference number | Author |
| --- | --- | --- |
| Deutsche Forschungsgemeinschaft | CRC 1310 | Joachim Krug |
| H2020 European Research Council | ERC Consolidator Grant 682237 EVOSTRUC | Rosalind J Allen |

The funders had no role in study design, data collection and interpretation, or the decision to submit the work for publication.

## Author contributions
Suman G Das, Conceptualization, Data curation, Software, Formal analysis, Investigation, Visualization, Writing - original draft, Writing - review and editing; Susana OL Direito, Resources, Data curation, Investigation, Writing - review and editing; Bartlomiej Waclaw, Conceptualization, Data curation, Supervision, Investigation, Writing - review and editing; Rosalind J Allen, Conceptualization, Supervision, Funding acquisition, Project administration, Writing - review and editing; Joachim Krug, Conceptualization, Formal analysis, Supervision, Investigation, Project administration, Writing - review and editing

## Author ORCIDs
Joachim Krug (iD) https://orcid.org/0000-0002-2143-6490

## Decision letter and Author response
Decision letter https://doi.org/10.7554/eLife.55155.sa1
Author response https://doi.org/10.7554/eLife.55155.sa2

# Additional files

## Supplementary files
• Transparent reporting form

## Data availability
Experimental data have been deposited in Edinburgh DataShare.

The following dataset was generated:

| Author(s) | Year | Dataset title | Dataset URL | Database and Identifier |
|---|---|---|---|---|
| Waclaw B, Direito S, Das S, Allen R, Krug J | 2020 | Experimental data for the article Predictable Properties of Fitness Landscapes Induced by Adaptational Tradeoffs | https://doi.org/10.7488/ds/2756 | Edinburgh DataShare, 10.7488/ds/2756 |

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
