## [Decision Letter]

**Acceptance summary:**

Models of fitness landscapes range from very simple peaks to unpredictable random spaces. In this manuscript, Das and colleagues describe a class of fitness landscapes that naturally emerge in drug resistance evolution, in which trade-offs result in rugged yet accessible and predictable landscapes.

**Decision letter after peer review:**

Thank you for submitting your article "Predictable Properties of Fitness Landscapes Induced by Adaptational Tradeoffs" for consideration by *eLife*. Your article has been reviewed by three peer reviewers, including Richard A Neher as the Reviewing Editor and Reviewer #1, and the evaluation has been overseen by Aleksandra Walczak as the Senior Editor. The following individual involved in review of your submission has agreed to reveal their identity: Weinreich (Reviewer #2). The reviewers have discussed the reviews with one another and the Reviewing Editor has drafted this decision to help you prepare a revised submission.

Summary:

Das et al. investigate the properties of fitness landscapes relevant for antimicrobial resistance evolution. They show that fitness peaks in these landscapes are accessible even though these landscapes are rugged. These results are obtained through elegant mathematical arguments that provide a clear intuition on why these landscapes are special yet natural. All reviewers felt that these results represent important progress in our understanding of fitness landscapes in general and antibiotic resistance evolution in particular.

Essential revisions:

1) Our main concern is the generality of the assumptions made in the analysis. We ask the authors to discuss available data that is compatible or that violates the assumptions. Ogbunugafor et al., 2016, for example analyze dose-response curves for Malaria drugs and show that high resistance mutants don't necessarily have lower fitness in the absence of the drug.

2) The multiplicative behavior of IC_50_ contributions is not obvious and the degree to which it holds might depend on the resistance mechanism. Would one expect similar multiplicative behavior for efflux pumps or drug-degrading enzymes? A general discussion of how the assumptions are compatible with different mechanisms of drug action and drug resistance would be welcome.

3) We encourage the authors to think of ways to provide intuitive insight into the structure of these fitness landscapes and connect to existing models and metaphors.

---

## [Author Response]

Essential revisions:1) Our main concern is the generality of the assumptions made in the analysis. We ask the authors to discuss available data that is compatible or that violates the assumptions. Ogbunugafor et al., 2016, for example analyze dose-response curves for Malaria drugs and show that high resistance mutants don't necessarily have lower fitness in the absence of the drug.

Our model rests on three key assumptions: (i) a tradeoff between growth and resistance, (ii) the scaling property of dose-response curves and (iii) the approximate absence of (positive) epistasis in null-fitness and IC_50_. The available data relevant to (ii) is rather limited, and we believe our discussion in the subsections “Scaling of dose-response curves” and “Limited epistasis in *r* and *m*”, is reasonably complete. Assumption (iii) is addressed in detail in the response to Essential revision 2) below. The reviewer’s reference to Ogbunugafor et al., 2016 concerns assumption (i). Here, there appears to be a broad consensus in the literature that tradeoffs are widespread (e.g., most resistance mutations come with a fitness cost) but exceptions in the form of compensatory mutations exist, and the data analyzed by Ogbunugafor et al. (originally obtained in Brown et al., 2010) belong to the latter category. We have added several references to the third paragraph of the Introduction where the tradeoff assumption is introduced, and extended the discussion of compensatory mutations in the third and eighth paragraphs of the Discussion. We also cite Brown et al., 2010 in this context.

2) The multiplicative behavior of IC_50_ contributions is not obvious and the degree to which it holds might depend on the resistance mechanism. Would one expect similar multiplicative behavior for efflux pumps or drug-degrading enzymes? A general discussion of how the assumptions are compatible with different mechanisms of drug action and drug resistance would be welcome.

We agree that this behaviour is not obvious. We have added two paragraphs to the Discussion that describe in detail possible scenarios leading to a multiplicative behavior of the IC_50_, and apply them to the system at hand (Discussion, fifth and sixth paragraphs).

3) We encourage the authors to think of ways to provide intuitive insight into the structure of these fitness landscapes and connect to existing models and metaphors.We have added a new Figure 8 that schematically illustrates different paradigmatic fitness landscape structures and a new paragraph (Discussion, seventh paragraph) where this figure is described.